# On the Universal Approximability and Complexity Bounds of Quantized ReLU Neural Networks

**Yukun Ding[1], Jinglan Liu[1], Jinjun Xiong[2], Yiyu Shi[1]**
[1] University of Notre Dame
[2] IBM Thomas J. Watson Research Center
{yding5,jliu16,yshi4}@nd.edu, jinjun@us.ibm.com

## Abstract

Compression is a key step to deploy large neural networks on resource-constrained platforms. As a popular compression technique, quantization constrains the number of distinct weight values and thus reducing the number of bits required to represent and store each weight. In this paper, we study the representation power of quantized neural networks. First, we prove the universal approximability of quantized ReLU networks on a wide class of functions. Then we provide upper bounds on the number of weights and the memory size for a given approximation error bound and the bit-width of weights for function-independent and function-dependent structures. Our results reveal that, to attain an approximation error bound of $\epsilon$, the number of weights needed by a quantized network is no more than $\mathcal{O}\left(\log^5(1/\epsilon)\right)$ times that of an unquantized network. This overhead is of much lower order than the lower bound of the number of weights needed for the error bound, supporting the empirical success of various quantization techniques. To the best of our knowledge, this is the first in-depth study on the complexity bounds of quantized neural networks.

## 1 Introduction

Various deep neural networks deliver state-of-the-art performance on many tasks such as object recognition and natural language processing using new learning strategies and architectures (Chen et al., 2017; He et al., 2016; Kumar et al., 2016; Ioffe & Szegedy, 2015; Vaswani et al., 2017). Their prevalence has extended to embedded or mobile devices for edge intelligence, where security, reliability or latency constraints refrain the networks from running on servers or in clouds. However, large network sizes with the associated expensive computation and memory consumption make edge intelligence even more challenging (Cheng et al., 2018; Sandler et al., 2018).

In response, as will be more detailed in Section 2, substantial effort has been made to reduce the memory consumption of neural networks while minimizing the accuracy loss. The memory consumption of neural networks can be reduced by either directly reducing the number of weights or decreasing the number of bits (bit-width) needed to represent and store each weight, which can be employed on top of each other (Choi et al., 2016). The number of weights can be reduced by pruning (Han et al., 2015b), weight sparsifying (Liu et al., 2015), structured sparsity learning (Wen et al., 2016) and low rank approximation (Denton et al., 2014). The bit-width is reduced by quantization that maps data to a smaller set of distinct levels (Sze et al., 2017). Note that while quantization may stand for linear quantization only (Li et al., 2017; Gysel et al., 2016) or nonlinear quantization only (Han et al., 2015a; Choi et al., 2016) in different works, our discussion will cover both cases.

However, as of today quantization is still only empirically shown to be robust and effective to compress various neural network architectures (Hubara et al., 2016; Zhou et al., 2017b; Zhuang et al., 2017). Its theoretical foundation still remains mostly missing. Specifically, many important questions remain unanswered. For example:

- Why even binarized networks, those most extremely quantized with bit-width down to one, still work well in some cases?

- To what extent will quantization decrease the expressive power of a network? Alternatively, what is the overhead induced by weight quantization in order to maintain the same accuracy?

In this paper, we provide some insights into these questions from a theoretical perspective. We focus on ReLU networks, which is among the most widely used in deep neural networks (Xu et al., 2015). We follow the idea from Yarotsky (2017) to prove the complexity bound by constructing a network, but with new and additional construction components essential for quantized networks. Specifically, given the number of distinct weight values $\lambda$ and a target function $f$, we construct a network that can approximate $f$ with an arbitrarily small error bound $\epsilon$ to prove the universal approximability. The memory size of this network then naturally serves as an upper bound for the minimal network size.

The high-level idea of our approach is to replace basic units in an unquantized network with quantized sub-networks [1] that approximate these basic units. For example, we can approximate a connection with any weight in an unquantized network by a quantized sub-network that only uses a finite number of given weight values. Even though the approximation of a single unit can be made arbitrarily accurate in principle with unlimited resources (such as increased network depth), in practice, there exists some inevitable residual error at every approximation, all of which could propagate throughout the entire network. The challenge becomes, however, how to mathematically prove that we can still achieve the end-to-end arbitrary small error bound even if these unavoidable residual errors caused by quantization can be propagated throughout the entire network. This paper finds a solution to solve the above challenge. In doing so, we have to propose a number of new ideas to solve related challenges, including judiciously choosing the proper finite weight values, constructing the approximation sub-networks as efficient as possible (to have a tight upper bound), and striking a good balance among the complexities of different approximation steps.

Based on the bounds derived, we compare them with the available results on unquantized neural networks and discuss its implications. In particular, the main contributions of this paper include:

- We prove that even the most extremely quantized ReLU networks using two distinct weight values are capable of representing a wide class of functions with arbitrary accuracy.

- Given the number of distinct weights and the desired approximation error bound, we provide upper bounds on the number of weights and the memory size. We further show that our upper bounds have good tightness by comparing them with the lower bound of unquantized ReLU networks established in the literature.

- We show that, to attain the same approximation error bound $\epsilon$, the number of weights needed by a quantized network is no more than $\mathcal{O}\left(\log^5(1/\epsilon)\right)$ times that of an unquantized network. This overhead is of much lower order compared with even the lower bound of the number of weights needed for the error bound. This partially explains why many state-of-the-art quantization schemes work well in practice.

- We demonstrate how a theoretical complexity bound can be used to estimate an optimal bit-width, which in turn enables the best cost-effectiveness for a given task.

The remainder of the paper is organized as follows. Section 2 reviews related works. Section 3 lays down the models and assumptions of our analysis. We prove the universal approximability and the upper bounds with function-independent structure in Section 4 and extend it to function-dependent structure in Section 5. We analyze the bound-based optimal bit-width in Section 6. Finally, Section 7 discusses the results and gets back to the questions raised above.

## 2 RELATED WORKS

**Quantized Neural Networks:** There are rich literatures on how to obtain quantized networks, either by linear quantization or nonlinear quantization (Zhou et al., 2017a; Leng et al., 2017; Shayar et al.,

---

[1]Throughout this paper, we use "sub-network" to denote a network that is used as a part of the final network that approximates a target function.

2017). Linear quantization does mapping with a same distance between contiguous quantization levels and is usually implemented by storing weights as fixed-point numbers with reduced bit-width (Li et al., 2017; Gysel et al., 2016). Nonlinear quantization maps the data to quantization levels that are not uniformly distributed and can be either preselected or learned from training. Then the weights are stored using lossless binary coding (the index to a lookup table) instead of the actual values (Han et al., 2015a; Choi et al., 2016). It is reported that a pruned AlexNet can be quantized to eight bits and five bits in convolutional layers and fully connected layers, respectively, without any loss of accuracy. Similar results are also observed in LENET-300-100, LENET-5, and VGG-16 (Han et al., 2015a). One may argue that some of these benchmark networks are known to have redundancy. However, recent works show that quantization works well even on networks that are designed to be extremely small and compact. SqueezeNet, which is a state-of-the-art compact network, can be quantized to 8-bit while preserving the original accuracy (Gysel et al., 2016; Iandola et al., 2016). There are some representative works that can achieve little accuracy loss on ImageNet classification even using binary or ternary weights (Courbariaux et al., 2015; Rastegari et al., 2016; Li et al., 2016; Zhu et al., 2016). More aggressively, some works also reduce the precision of activations, e.g. (Hubara et al., 2016; Rastegari et al., 2016; Faraone et al., 2018). Although the classification accuracy loss can be minimized, the universal approximation property is apparently lost, as with limited output precision the network cannot achieve arbitrary accuracy. Accordingly, we do not include them in the discussion of this paper. The limit of quantization is still unknown while the state-of-the-art keeps getting updated. For example, VGG-16 is quantized to 3-bit while maintaining the original accuracy (Leng et al., 2017). Motivated by the great empirical success, the training of quantized neural networks has been analyzed theoretically, but not the network capacity (Li et al., 2017; Choi et al., 2016).

**Universal Approximability and Complexity Bounds:** The universal approximability of ReLU networks is proved in Mhaskar & Micchelli (1992) and revisited in (Sonoda & Murata, 2017). Recently, Hanin (2017) discusses the expressive power of ReLU networks with bounded width and proves that a ReLU network with width $d+1$ can approximate any continuous convex function of $d$ variables arbitrarily well. Shaham et al. (2016) construct a sparsely-connected depth-4 ReLU network and prove its error bound. Liang & Srikant (2016) prove that, for a large class of piecewise smooth functions, a network with ReLU and binary step units can provide an error bound $\epsilon$ with $\mathcal{O}(1/\epsilon)$ layers and $\mathcal{O}(\text{poly} \log(1/\epsilon))$ neurons. The universal approximation property of low displacement rank (LDR) neural networks has been proved by Zhao et al. (2017) under a mild condition on the displacement operator, which is the result of another effective technique of neural network compression.

## 3 MODELS AND ASSUMPTIONS

Throughout this paper, we define ReLU networks as feedforward neural networks with the ReLU activation function $\sigma(x) = \max(0, x)$. The ReLU network considered includes multiple input units, a number of hidden units, and one output unit. Without loss of generality, each unit can only connect to units in the next layer. Our conclusions on ReLU networks can be extended to any other networks that use piecewise linear activation functions with finite breakpoints such as leaky ReLU and ReLU-6 immediately, as one can replace a ReLU network by an equivalent one using these activation functions while only increasing the number of units and weights by constant factors (Yarotsky, 2017).

We denote the finite number of distinct weight values as $\lambda$ ($\lambda \in \mathbb{Z}^+$ and $\lambda \geq 2$), for both linear quantization and nonlinear quantization. For linear quantization, without loss of generality, we assume the finite number of distinct weight values are given as $\{-1, \frac{1}{\lambda}, \frac{2}{\lambda}, \ldots, \frac{\lambda-1}{\lambda}\}$, where $\{\frac{1}{\lambda}, \frac{2}{\lambda}, \ldots, \frac{\lambda-1}{\lambda}\}$ are uniformly spaced (hence called "linear") in $(0, 1)$ and $-1$ is used to obtain the negative weight values. For nonlinear quantization, we assume the finite number of distinct weight values are not constrained to any specific values, i.e., they can take any values as needed. To store each weight, we only need $\log(\lambda)$ [2] bits to encode the index, i.e. the bit-width is $\log(\lambda)$. The overhead to store sparse structures can be ignored because it varies depending on the implementation and can be easily reduced to the same order as the weight storage using techniques such as compressed sparse row (CSR) for nonlinear quantization. The number of bits needed to store the codebook can also be ignored because it has lower order of complexity.

---

[2]Throughout this paper, we omit base 2 for clarity of presentation

We consider any function $f$ in the Sobolev space: $f \in \mathcal{W}^{n,\infty}([0,1]^d)$ and

$$\max_{\mathbf{n}:|\mathbf{n}|\le n} \operatorname{ess\,sup}_{x\in[0,1]^d} |D^{\mathbf{n}}f(\mathbf{x})| \le 1. \tag{1}$$

The space $\mathcal{W}^{n,\infty}$ consists of all locally integrable function $f : \Omega \to \mathbb{R}$ such that $D^{\mathbf{n}}f \in L^{\infty}(\Omega)$, where $|\mathbf{n}| \le n$ and $\Omega$ is an open set in $\mathbb{R}^d$. We denote this function space as $\mathcal{F}_{d,n}$ in this paper.

Note that we only assume weak derivatives up to order $n$ exist where $n$ can be as small as 1 where the function is non-differentiable. We also only assume the Lipschitz constant to be no greater than 1 for the simplicity of the derivation. When the Lipschitz constant is bigger than 1, as long as it is bounded, the whole flow of the proof remains the same though the bound expression will scale accordingly.

When constructing the network to approximate any target function $f$, we consider two scenarios for deriving the bounds. The first scenario is called function-dependent structure, where the constructed network topology and their associated weights are all affected by the choice of the target function. In contrast, the second scenario is called function-independent structure, where the constructed network topology is independent of the choice of the target function in $f \in \mathcal{F}_{d,n}$ with a given $\epsilon$. The principle behind these design choices (the network topology constructions and the choice of weights) is to achieve a tight upper bound as much as possible.

One might consider that we can transform an unquantized network within the error bound to a quantized one in a straightforward way by approximating every continuous-value weight with a combination of discrete weights with arbitrary accuracy. However, the complexity of such approximation (number of discrete weights needed) depends on the distribution of those continuous-value weights (e.g., their min and max), which may vary depending on the training data and network structure and a closed-form expression for the upper bounds is not possible. As such, a more elegant approach is needed. Below we will establish a constructive approach which allows us to bound the approximation analytically.

## 4 Function-independent Structure

We start our analysis with function-independent structure, where the network topology is fixed for any $f \in \mathcal{F}_{d,n}$ and a given $\epsilon$. We first present the approximation of some basic functions by sub-networks in Section 4.1. We then present the sub-network that approximates any weight in Section 4.2, and finally the approximation of general functions and our main results are in Section 4.3.

### 4.1 Approximation of squaring/multiplication

**Proposition 1.** *Denote the design parameter that determines the approximation error bound as $r$. Let $f_s^r$ be a ReLU sub-network with only two weight values $\frac{1}{2}$ and $-\frac{1}{2}$. The function $f_s(x) = x^2$ on the segment $[0,1]$ can be approximated by $f_s^r$, such that (i) if $x = 0$, $f_s^r(x) = 0$; (ii) the approximation error $\epsilon_s \le 2^{-2(r+1)}$; (iii) the depth is $\mathcal{O}(r)$; (iv) the width is a constant; (v) the number of weight is $\mathcal{O}(r)$.*

The proof and the details of the sub-network constructed are included in Appendix A.1. Once the approximation to squaring function is obtained, we get Proposition 2 by the fact that $2xy = (x+y)^2 - x^2 - y^2$.

**Proposition 2.** *Denote the design parameter that determines the approximation error bound as $r$. Given $x \in [-1,1]$, $y \in [-1,1]$, and only two weight values $\frac{1}{2}$ and $-\frac{1}{2}$, there is a ReLU sub-network with two input units that implements a function $\times': \mathbb{R}^2 \mapsto \mathbb{R}$, such that (i) if $x = 0$ or $y = 0$, then $\times'(x,y) = 0$; (ii) for any $x$, $y$, the error $\epsilon_{\times'} = |\times'(x,y) - xy| \le 6 \cdot 2^{-2(r+1)}$; (iii) the depth is $\mathcal{O}(r)$; (iv) the width is a constant; (v) the number of weights is $\mathcal{O}(r)$.*

*Proof.* Build three sub-networks $f_s^r$ as described in Proposition 1 and let

$$\times'(x,y) = 2\left(f_s^r(|x+y|/2) - f_s^r(|x|/2) - f_s^r(|y|/2)\right). \tag{2}$$

Then the statement (i) is followed by property (i) of Proposition 1. Using the error bound in Proposition 1 and Equation (2), we get the error bound of $\times'$:

$$\epsilon_{\times'} \le 6 \cdot 2^{-2(r+1)}. \tag{3}$$

Since a sub-network $B_{abs}$ that computes $\sigma(x) + \sigma(-x)$ can be constructed to get the absolute value of $x$ trivially, we can construct $\times'(x, y)$ as a linear combination of three parallel $f_s^r$ and feed them with $\frac{|x|}{2}, \frac{|y|}{2}$, and $\frac{|x+y|}{2}$. Then claims of statement (iii), (iv), and (v) are also obtained. □

## 4.2 Approximation of weights

**Proposition 3.** *Denote the design parameter that determines the approximation error bound as $t$. A connection with any weight $w \in [-1, 1]$ can be approximated by a ReLU sub-network that has only $\lambda \geq 2$ distinct weights, such that (i) the sub-network is equivalent to a connection with weight $w'$ while the approximation error is bounded by $2^{-t}$ i.e., $|w' - w| < 2^{-t}$; (ii) the depth is $\mathcal{O}\left(\lambda t^{\frac{1}{\lambda-1}}\right)$; (iii) the width is $\mathcal{O}(t)$; (iv) the number of weights is $\mathcal{O}\left(\lambda t^{\frac{1}{\lambda-1}+1}\right)$.*

*Proof.* Consider that we need a weight $w$ to feed the input $x$ to a unit in the next layer as $wx$. With a limited number of distinct weight values, we can construct the weight we need by cascade and combination.

For clarity, we first consider $w \geq 0$ and $x \geq 0$, and relax these assumptions later. The connections with $w = 0$ can be seen as an empty sub-network while $w = 1$ can be easily implemented by 4 units with weight $\frac{1}{2}$. Now we show how to represent all integral multiples of $2^{-t}$ from $2^{-t}$ to $1 - 2^{-t}$, which will lead to the statement (i) by choosing the nearest one from $w$ as $w'$. Without loss of generality, we assume $t^{\frac{1}{\lambda-1}}$ is an integer. We use $\lambda$ weights that include $-\frac{1}{2}$ and $W$:

$$W \triangleq \{2^{-1}, 2^{-t^{\frac{1}{\lambda-1}}}, 2^{-t^{\frac{2}{\lambda-1}}}, \cdots, 2^{-t^{\frac{\lambda-2}{\lambda-1}}}\}. \tag{4}$$

We first construct all $w$ from $W_c$ which is defined as

$$W_c \triangleq \{2^{-1}, 2^{-2}, \cdots, 2^{-(t-1)}\}. \tag{5}$$

Similar to a numeral system with radix equal to $t^{\frac{1}{\lambda-1}}$, any $w_i \in W_c$ can be obtained by concatenating weights from $W$ while every weights in $W$ is used no greater than $t^{\frac{1}{\lambda-1}} - 1$ times.

After that, all integral multiples of $2^{-t}$ from $2^{-t}$ to $1 - 2^{-t}$ can be represented by a binary expansion on $W_c$. Note that connections in the last layer for binary expansion use weight $\frac{1}{2}$, thus additional $2^{-1}$ is multiplied to scale the resolution from $2^{-(t-1)}$ to $2^{-t}$. Since for any weight in $W_c$ we need to concatenate no more than $\lambda\left(t^{\frac{1}{\lambda-1}} - 1\right)$ weights in a straight line, the sub-network has no greater than $\lambda\left(t^{\frac{1}{\lambda-1}} - 1\right) + 1$ layers, and no greater than $4t\lambda\left(t^{\frac{1}{\lambda-1}} - 1\right) + 8t + 4$ weights.

We now relax the assumption $w \geq 0$. When $w < 0$, the sub-network can be constructed as $w = |w|$, while we use $-\frac{1}{2}$ instead of $\frac{1}{2}$ in the last layer. To relax the assumption $x \geq 0$, we can make a duplication of the sub-network. Let all the weights in the first layer of the sub-network be $\frac{1}{2}$ for one and $-\frac{1}{2}$ for the other. Here we are utilizing the gate property of ReLU. In this way, one sub-network is activated only when $x > 0$ and the other is activated only when $x < 0$. The sign of the output can be adjusted by flipping the sign of weights in the last layer. Note that the configuration of the sub-network is solely determined by $w$ and works for any input $x$. □

The efficiency of the weight approximation is critical to the overall complexity. Compared with the weight selection as $\{2^{-1}, 2^{-t^{\frac{1}{\lambda-1}}}, 2^{-t^{\frac{2}{\lambda-1}}}, \ldots, 2^{-t^{\frac{(\lambda-2)}{\lambda-1}}}\}$, our approximation reduces the number of weights by a factor of $t^{\frac{\lambda-2}{\lambda-1}}$.

## 4.3 Approximation of general functions

With the help of Proposition 2 and Proposition 3, we are able to prove the upper bound for general functions.

**Theorem 1.** *For any $f \in \mathcal{F}_{d,n}$, given $\lambda$ distinct weights, there is a ReLU network with fixed structure that can approximate $f$ with any error $\epsilon \in (0, 1)$, such that (i) the depth is*

$\mathcal{O}\left(\lambda \log^{\frac{1}{\lambda-1}}(1/\epsilon) + \log(1/\epsilon)\right)$; *(ii) the number of weights is* $\mathcal{O}\left(\lambda \log^{\frac{1}{\lambda-1}+1}(1/\epsilon)(1/\epsilon)^{\frac{d}{n}}\right)$; *(iii) the number of bits needed to store the network is* $\mathcal{O}\left(\lambda \log(\lambda) \log^{\frac{1}{\lambda-1}+1}(1/\epsilon)(1/\epsilon)^{\frac{d}{n}}\right)$.

The complete proof and the network constructed can be found in Appendix A.2. We first approximate $f$ by $f_2$ using the Taylor polynomial of order $n-1$ and prove the approximation error bound. Note that even when $f$ is non-differentiable (only first order weak derivative exists), the Taylor polynomial of order 0 at $\mathbf{x} = \frac{\mathbf{m}}{N}$ can still be used, which takes the form of $P_{\mathbf{m}} = f(\frac{\mathbf{m}}{N})$. Then we approximate $f_2$ by a ReLU network that is denoted as $f'$ with bounded error. After that, we present the quantized ReLU network that implements $f'$ and the complexity.

The discussion above focuses on nonlinear quantization which is a more general case compared to linear quantization. For linear quantization, which strictly determines the available weight values once $\lambda$ is given, we can use the same proof for nonlinear quantization except for a different subnetwork for weight approximation with width $t$ and depth $\frac{t}{\log \lambda}+1$. Here we give the theorem and the proof is included in Appendix A.3.

**Theorem 2.** *For any $f \in \mathcal{F}_{d,n}$, given weight maximum precision $\frac{1}{\lambda}$, there is a ReLU network with fixed structure that can approximate $f$ with any error $\epsilon \in (0,1)$, such that (i) the depth is* $\mathcal{O}(\log(1/\epsilon))$; *(ii) the number of weights is* $\mathcal{O}\left(\left(\log(1/\epsilon) + \frac{\log^2(1/\epsilon)}{\log \lambda}\right)(1/\epsilon)^{\frac{d}{n}}\right)$; *(iii) the number of bits needed to store the network is* $\mathcal{O}\left(\left(\log(\lambda)\log(1/\epsilon) + \log^2(1/\epsilon)\right)(1/\epsilon)^{\frac{d}{n}}\right)$.

## 5 FUNCTION-DEPENDENT STRUCTURE

The network complexity can be reduced if the network topology can be set according to a specific target function, i.e. function-dependent structure. In this section, we provide an upper bound for function-dependent structure when $d = 1$ and $n = 1$, which is asymptotically better than that of a fixed structure. Specifically, we first define an approximation to $f(x)$ as $\widetilde{f}(x)$ that has special properties to match the peculiarity of quantized networks. Then we use piecewise linear interpolation and "cached" functions (Yarotsky, 2017) to approximate $\widetilde{f}(x)$ by a ReLU network.

### 5.1 FUNCTION TRANSFORMATION

While simply using piecewise linear interpolation at the scale of $\epsilon$ can satisfy the error bound with $\mathcal{O}(1/\epsilon)$ weights, the complexity can be reduced by first doing interpolation at a coarser scale and then fill the details in the intervals to make the error go down to $\epsilon$. By assigning a "cached" function to every interval depending on specific function and proper scaling, the number of weights is reduced to $\mathcal{O}\left(\left(\log^{-1}(1/\epsilon)\right)1/\epsilon\right)$ when there is no constraint on weight values (Yarotsky, 2017).

The key difficulty in applying this approach to quantized ReLU networks is that the required linear interpolation at $\frac{i}{T}$ exactly where $i = 1, 2, \cdots, T$ is not feasible because of the constraint on weight selection. To this end, we transform $f(x)$ to $\widetilde{f}(x)$ such that the approximation error is bounded; the Lipschitz constant is preserved; $\widetilde{f}\left(\frac{i}{T}\right)$ are reachable for the network under the constraints of weight selection without increasing the requirement on weight precision. Then we can apply the interpolation and cached function method on $\widetilde{f}(x)$ and finally approximate $f(x)$ with a quantized ReLU network. Formally, we get the following proposition and the proof can be found in Appendix A.4.

**Proposition 4.** *For any $f \in \mathcal{F}_{1,1}$, $t \in \mathbb{Z}^+$, and $T \in \mathbb{Z}^+$, there exists a function $\widetilde{f}(x)$ such that (i) $\widetilde{f}(x)$ is a continuous function with Lipschitz constant 1; (ii) $\widetilde{f}(\frac{i}{T}) = \left\lceil Tf\left(\frac{i}{T}\right)/2^{-t}\right\rceil \frac{2^{-t}}{T}$; (iii) $|\widetilde{f}(x) - f(x)| < \frac{2^{-t}}{T}$.*

### 5.2 APPROXIMATION BY RELU NETWORKS

With the help of Proposition 4 and the weight construction method described in Section 4.2, we are able to apply the interpolation and cached function approach. Denoting the output of the network as $f''(x)$, we have $|f(x) - f''(x)| = |f(x) - \widetilde{f}(x)| + |\widetilde{f}(x) - f''(x)| \leq \epsilon$ by choosing appropriate hyperparameters which are detailed in Appendix A.5 and the network complexity is obtained accordingly.

**Theorem 3.** *For any $f \in \mathcal{F}_{1,1}$ , given $\lambda$ distinct weights, there is a ReLU network with function-dependent structure that can approximate $f$ with any error $\epsilon \in (0,1)$, such that (i) the depth is $\mathcal{O}\left(\lambda \left(\log \log \left(1/\epsilon\right)\right)^{\frac{1}{\lambda-1}} + \log \left(1/\epsilon\right)\right)$; (ii) the number of weights is $\mathcal{O}\left(\lambda \left(\log \log \left(1/\epsilon\right)\right)^{\frac{1}{\lambda-1}+1} + (1/\epsilon)\right)$ (iii) the number of bits needed to store the network is $\mathcal{O}\left(\log \lambda \left(\lambda \left(\log \log \left(1/\epsilon\right)\right)^{\frac{1}{\lambda-1}+1} + (1/\epsilon)\right)\right)$.*

Using the different weight construction approach as in the case of function-independent structure, we have the result for linear quantization:

**Theorem 4.** *For any $f \in \mathcal{F}_{1,1}$ , given weight maximum precision $\frac{1}{\lambda}$, there is a ReLU network with function-dependent structure that can approximate $f$ with any error $\epsilon \in (0,1)$, such that (i) the depth is $\mathcal{O}\left(\log \left(1/\epsilon\right)\right)$; (ii) the number of weights is $\mathcal{O}\left(1/\epsilon\right)$; (iii) the number of bits needed to store the network is $\mathcal{O}\left(\log(\lambda)/\epsilon\right)$.*

## 6 BOUND-BASED OPTIMAL BIT-WIDTH

In this section, we first introduce the optimal bit-width problem and then show how a theoretical bound could potentially be used to estimate the optimal bit-width of a neural network.

Because of the natural need and desire of comparison with competitive approaches, most quantization techniques are evaluated on some popular reference networks, without modification of the network topology. On the one hand, the advancement of lossless quantization almost stalls at a bit-width between two and six (Han et al., 2015a; Choi et al., 2016; Sze et al., 2017; Blott et al., 2017; Su et al., 2018; Faraone et al., 2018). A specific bit-width depends on the compactness of the reference network and the difficulty of the task. On the other hand, the design space, especially the different combinations of topology and bit-width, is largely underexplored because of the complexity, resulting in sub-optimal results. A recent work by Su et al. (2018) empirically validates the benefit of exploring flexible network topology during quantization. That work adds a simple variable of network expanding ratio, and shows that a bit-width of four achieves the best cost-accuracy trade-off among limited options in $\{1, 2, 4, 8, 16, 32\}$. Some recent effort on using reinforcement learning to optimize the network hyper-parameters (He et al., 2018) could potentially be used to address this issue. But the current design space is still limited to a single variable per layer (such as the pruning ratio based on a reference network). How to estimate an optimal bit-width for a target task without training could be an interesting research direction in the future.

The memory bound expression as derived in this paper helps us to determine whether there is an optimal $\lambda$ that would lead to the lowest bound and most compact network (which can be translated to computation cost in a fully connected structure) for a given target function. For example, by dropping the lower-order term and ignoring the rounding operator, our memory bound can be simplified as

$$M(\lambda) = \theta_1 \lambda \log(\lambda) \log^{\frac{1}{\lambda-1}+1}(3n2^d/\epsilon) \tag{6}$$

where $\theta_1$ is a constant determined by $\epsilon$, $n$, and $d$. We can find an optimal $\lambda$ that minimizes $M(\lambda)$:

$$\lambda_{opt} = \underset{\lambda}{\operatorname{argmin}} M(\lambda) \tag{7}$$

As is detailed in Appendix B, we prove that there exists one and only one local minimum (hence global minimum) in the range of $[2, \infty)$ whenever $\epsilon < \frac{1}{2}$. We also show that $\lambda_{opt}$ is determined by $\log\left(3n2^d/\epsilon\right)$, which can be easily dominated by $d$. Based on such results, we quantitatively evaluate the derivative of $M(\lambda)$, and based on which the optimal bit-width $\log(\lambda_{opt})$ under various settings in Figure 1a and Figure 1b, respectively. In Figure 1b, we also mark the input dimension of a few image data sets. It is apparent to see that the optimal bit width derived from $M(\lambda)$ is dominated by $d$ and lies between one and four for a wide range of input size. This observation is consistent with most existing empirical research results, hence showing the potential power of our theoretical bound derivation.

Since the bounds are derived for fully connected networks and depend on the construction approach, the interesting proximity between $\log(\lambda_{opt})$ and the empirical results cannot be viewed as a strict theoretical explanation. Regardless, we show that the complexity bound may be a viable approach

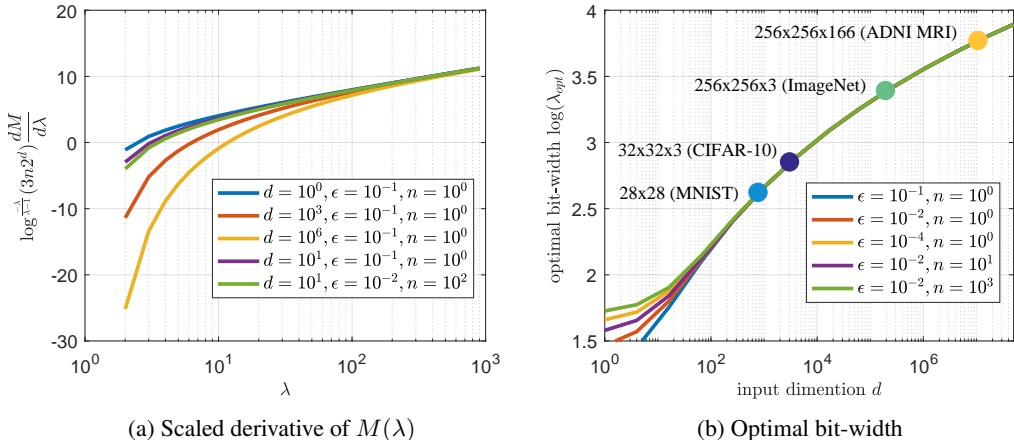

(a) Scaled derivative of $M(\lambda)$          (b) Optimal bit-width

Figure 1: Quantitative evaluation of the derivative of $M(\lambda)$ and the optimal bit-width $\log(\lambda_{opt})$. The derivative is scaled by $\log^{\frac{-\lambda}{\lambda-1}}(3n2^d)$ to fit in the same range. $\log^{\frac{-\lambda}{\lambda-1}}(3n2^d)$ is a positive monotonically increasing function and thus does not affect the trends too much. Note that $\lambda$ is the number of distinct weight values and thus $\log(\lambda)$ is the corresponding bit-width. It can be seen that $\epsilon$ and $n$ only affect $\log(\lambda_{opt})$ when $d$ is small ($< 10^2$). We also mark the input dimension $d$ of various image data set and their corresponding $\log(\lambda_{opt})$. It shows that the optimal bit-width increases very slowly with $d$.

to understand the optimal bit-width problem, thus potentially accelerating the hyper-parameter optimization of deep neural networks. We defer such a thorough investigation of the optimal bit-width or optimal hybrid bit-width configuration across the network to our future work.

## 7 DISCUSSION

In this section, we further discuss the bound of nonlinear quantization with a function-independent structure as the generality of nonlinear quantization. The availability of unquantized function-independent structures in literature also makes it an excellent reference for comparison.

**Comparison with the Upper Bound:** The quality of an upper bound lies on its tightness. Compared with the most recent work on unquantized ReLU networks (Yarotsky, 2017), where the upper bound on the number of weights to attain an approximation error $\epsilon$ is given by $\mathcal{O}\left(\log(1/\epsilon)\left(1/\epsilon\right)^{\frac{d}{n}}\right)$, our result for a quantized ReLU network is given by $\mathcal{O}\left(\lambda\left(\log^{\frac{1}{\lambda-1}+1}(1/\epsilon)\right)\left(1/\epsilon\right)^{\frac{d}{n}}\right)$, which translates to an increase by a factor of $\lambda\left(\log^{\frac{1}{\lambda-1}}(1/\epsilon)\right)$. Loosely speaking, this term reflects the loss of expressive power because of weight quantization, which decreases quickly as $\lambda$ increases.

**Comparison with the Lower Bound:** We also compare our bound with the lower bound of the number of weights needed to attain an error bound of $\epsilon$ to have a better understanding on the tightness of the bound. We use the lower bound for unquantized ReLU networks from (Yarotsky, 2017), as it is also a natural lower bound for quantized ReLU networks. Under the same growth rate of depth, the lower bound is given by $\Omega(\log^{-3}(1/\epsilon)\left(1/\epsilon\right)^{d/n})$, while our upper bound is, within a polylog factor when $\lambda$ is a constant, $\mathcal{O}(\lambda\log^{\frac{1}{\lambda-1}+1}(1/\epsilon)(1/\epsilon)^{d/n})$. The comparison validates the good tightness of our upper bound.

**The Upper Bound of Overhead:** More importantly, the above comparison yields an upper bound on the possible overhead induced by quantization. By comparing the expressions of two bounds while treating $\lambda$ as a constant, we can show that, to attain the same approximation error bound $\epsilon$, the number of weights needed by a quantized ReLU network is no more than $\mathcal{O}(\log^5(1/\epsilon))$ times that needed by an unquantized ReLU network. Note that this factor is of much lower order than the lower bound $\Omega(\log^{-3}(1/\epsilon)\left(1/\epsilon\right)^{d/n})$. This little overhead introduced by weight quantization explains in part the empirical success on network compression and acceleration by quantization and

also answers in part the questions as raised in Section 1. Given the significant benefits of quantization in term of memory and computation efficiency, we anticipate that the use of quantization networks will continue to grow, especially on resource-limited platforms.

**Future Work:** There remain many other avenues for future investigation. For example, although we derived the first upper bound of quantized neural networks, the lower bound is still missing. If a tight lower bound of the network size is established, it could be combined with the upper bound to give a much better estimation of required resources and the optimal bit-width. We believe the trends associated with the bounds can also be useful and deserve some further investigation. For example, the trend may help hardware designers in their early stage of design exploration without the need of lengthy training. While we assume a uniform bit-width across all layers, another area of research is to allow different bit-widths in different layers, which could achieve better efficiency and potentially provide theoretical justifications on the emerging trend of hybrid quantization (Zhang et al., 2017; Wang et al., 2018).

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

## A    PROOFS

### A.1    THE PROOF OF PROPOSITION 1

**Proposition 1.** *Denote the design parameter that determines the approximation error bound as $r$. Let $f_s^r$ be a ReLU sub-network with only two weight values $\frac{1}{2}$ and $-\frac{1}{2}$. The function $f_s(x) = x^2$ on the segment $[0,1]$ can be approximated by $f_s^r$, such that (i) if $x = 0$, $f_s^r(x) = 0$; (ii) the approximation error $\epsilon_s \leq 2^{-2(r+1)}$; (iii) the depth is $\mathcal{O}(r)$; (iv) the width is a constant; (v) the number of weight is $\mathcal{O}(r)$.*

*Proof.* For $f_s(x) = x^2$, let $f_s^r$ be the piecewise linear interpolation of $f$ with $2^r + 1$ uniformly distributed breakpoints $\frac{k}{2^r}, k = 0, \dots, 2^r$. We have $f_s^r\left(\frac{k}{2^r}\right) = \left(\frac{k}{2^r}\right)^2, k = 0, \dots, 2^r$ and the approximation error $\epsilon_s = ||f_s^r(x) - f_s(x)||_\infty \leq 2^{-2(r+1)}$. We can use the function $g : [0,1] \mapsto [0,1]$ to obtain $f_s^r$:

$$g(x) = \begin{cases} 2x & x < \frac{1}{2} \\ 2(1-x) & x \geq \frac{1}{2}, \end{cases} \tag{8}$$

$$f_s^r(x) = x - \sum_{i=1}^{r} 2^{-2i} g^{\circ i}(x) \tag{9}$$

where $g^{\circ i}(x)$ is the $i$-th iterate of $g(x)$. Since $g(x)$ can be implemented by a ReLU sub-network as $g(x) = 2\sigma(x) - 4\sigma(x - \frac{1}{2})$, $g^{\circ r}(x)$ can be obtained by concatenating such implementation of $g(x)$ for $r$ times. Now, to implement $f_s^r(x)$ based on $g^{\circ r}(x)$, all we need are weights $\{2^{-2}, 2^{-4}, \cdots, 2^{-2(r-1)}, 2^{-2r}\}$, which can be easily constructed with additional $2r$ layers and the weight $\frac{1}{2}$.

Note that a straightforward implementation will have to scale $g^{\circ i}(x)$ separately (multiply by different numbers of $\frac{1}{2}$) before subtracting them from $x$ because each $g^{\circ i}(x)$ have a different coefficient. Then the width of the network will be $\Theta(r)$. Here we use a "pre-scale" method to reduce the network width from $\Theta(r)$ to a constant. The network constructed is shown in Figure 2. The one-layer sub-network that implements $g(x)$ and the one-layer sub-network that scales the input by $4$ are denoted as $B_g$ and $B_m$ respectively. Some units are copied to compensate the scaling caused by $\frac{1}{2}$. The intermediate results $g^{\circ i}(x)$ are computed by the concatenation of $B_g$ at the $(i+1)$-th layer. The first $B_m$ takes $x$ as input and multiply it by $4$. The output of $i$-th $B_m$ is subtracted by $g^{\circ i}(x)$ and then fed to the next $B_m$ to be multiplied by $4$ again. There are $r$ layers of $B_m$ and all $g^{\circ i}(x)$ are scaled by $2^{2(r-i)}$ respectively. As a result, we obtain $2^{2r}x - \sum_{i=1}^{r} 2^{2(r-i)} g^{\circ i}(x)$ after the last $B_m$. Then it is scaled by $2^{-2r}$ in the later $2r$ layers to get $f_s^r(x)$. In this way, we make all $g^{\circ i}(x)$ sharing the same scaling link and a constant width can be achieved.

$\square$

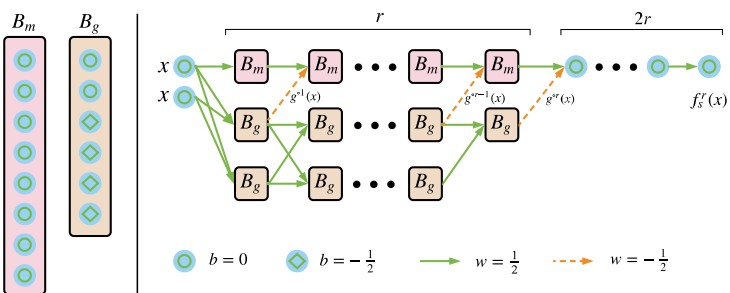

Figure 2: A qunatized ReLU network that implements $f_s^r(x)$. In the interest of clarity, we depict the sub-networks with different functions as different colored blocks. A connection from or to a block indicates connections from or to all units in the block. Details of block $B_m$ and block $B_g$ are depicted on the left. $b$ and $w$ denote bias and weight respectively.

## A.2 THE PROOF OF THEOREM 1

**Theorem 1.** *For any $f \in \mathcal{F}_{d,n}$, given $\lambda$ distinct weights, there is a ReLU network with fixed structure that can approximate $f$ with any error $\epsilon \in (0,1)$, such that (i) the depth is $\mathcal{O}\left(\lambda \log^{\frac{1}{\lambda-1}}(1/\epsilon) + \log(1/\epsilon)\right)$; (ii) the number of weights is $\mathcal{O}\left(\lambda \log^{\frac{1}{\lambda-1}+1}(1/\epsilon)(1/\epsilon)^{\frac{d}{n}}\right)$; (iii) the number of bits needed to store the network is $\mathcal{O}\left(\lambda \log(\lambda) \log^{\frac{1}{\lambda-1}+1}(1/\epsilon)(1/\epsilon)^{\frac{d}{n}}\right)$.*

*Proof.* The proof is composed of four steps. We first approximate $f$ by $f_2$ using the Taylor polynomial of order $n-1$ and prove the approximation error bound. Note that even when $f$ is non-differentiable (only first order weak derivative exist), the Taylor polynomial of order 0 at $\mathbf{x} = \frac{\mathbf{m}}{N}$ can still be used, which takes the form of $P_{\mathbf{m}} = f(\frac{\mathbf{m}}{N})$. Then we approximate $f_2$ by a ReLU network that is denoted as $f'$ with bounded error. After that, we present the quantized ReLU network that implements the network $f'$ and the complexity of the network.

We use a partition of unity on $[0,1]^d$: $\sum_{\mathbf{m}} \psi_{\mathbf{m}}(\mathbf{x}) \equiv 1, \mathbf{x} \in [0,1]^d$ where $\mathbf{m} = (m_1, \cdots, m_d) \in \{0, 1, \cdots, N\}^d$, and $h(x)$ is defined as follows:

$$\psi_{\mathbf{m}}(\mathbf{x}) = \prod_{k=1}^{d} h(3Nx_k - 3m_k), \tag{10}$$

where $N$ is a constant and

$$h(x) = \begin{cases} 1 & |x| \leq 1 \\ 2 - |x| & 1 < |x| < 2 \\ 0 & |x| \geq 2 \end{cases}. \tag{11}$$

Note that $\text{supp } \psi_{\mathbf{m}} \subset \{\mathbf{x} : \left|x_k - \frac{m_k}{N}\right| < \frac{1}{N} \forall k\}$. For all $\mathbf{m}$, we have the order $n-1$ Taylor polynomial for the function $f$ at $x = \frac{\mathbf{m}}{N}$ as

$$P_{\mathbf{m}}(\mathbf{x}) = \sum_{\mathbf{n}:|\mathbf{n}|<n} \frac{D^{\mathbf{n}}f}{\mathbf{n}!}\Big|_{\mathbf{x}=\frac{\mathbf{m}}{N}} \left(\mathbf{x} - \frac{\mathbf{m}}{N}\right)^{\mathbf{n}}. \tag{12}$$

To get a more realizable approximation for quantized networks, we define $P'_{\mathbf{m}}(\mathbf{x}) = \sum_{\mathbf{n}:|\mathbf{n}|<n} \beta_{\mathbf{m},\mathbf{n}} \left(\mathbf{x} - \frac{\mathbf{m}}{N}\right)^{\mathbf{n}}$ where $\beta_{\mathbf{m},\mathbf{n}}$ is $\frac{D^{\mathbf{n}}f}{\mathbf{n}!}\Big|_{\mathbf{x}=\frac{\mathbf{m}}{N}}$ rounded to the nearest integral multiple of $\frac{1}{n}\left(\frac{d}{N}\right)^{n-|\mathbf{n}|}$. Then we get an approximation to $f$ using $P'_{\mathbf{m}}$ and $\psi_{\mathbf{m}}$ as $f_2 \triangleq \sum_{\mathbf{m}\in\{0,\cdots,N\}^d} \psi_{\mathbf{m}} P'_{\mathbf{m}}$.

Then the approximation error of $f_2$ is bounded by Equation (13).

$$\begin{aligned}
|f(\mathbf{x}) - f_2(\mathbf{x})| &= |\sum_{\mathbf{m}} \psi_{\mathbf{m}}(\mathbf{x})(f(\mathbf{x}) - P'_{\mathbf{m}}(\mathbf{x}))| \\
&\leq \sum_{\mathbf{m}:|x_k-\frac{m_k}{N}|<\frac{1}{N}\forall k} |f(\mathbf{x}) - P'_{\mathbf{m}}(\mathbf{x})| \\
&\leq 2^d \max_{\mathbf{m}:|x_k-\frac{m_k}{N}|<\frac{1}{N}\forall k} |f(\mathbf{x}) - P_{\mathbf{m}}(\mathbf{x})| + 2^d \max_{\mathbf{m}:|x_k-\frac{m_k}{N}|<\frac{1}{N}\forall k} |P_{\mathbf{m}}(\mathbf{x}) - P'_{\mathbf{m}}(\mathbf{x})| \\
&\leq \frac{2^d d^n}{n!}\left(\frac{1}{N}\right)^n \max_{\mathbf{n}:|\mathbf{n}|=n}\operatorname*{ess\,sup}_{\mathbf{x}\in[0,1]^d} |D^{\mathbf{n}}f(\mathbf{x})| + 2^d \sum_{\mathbf{n}:|\mathbf{n}|<n} \max\left(|\beta_{\mathbf{m},\mathbf{n}} - \frac{D^{\mathbf{n}}f}{\mathbf{n}!}|\right)(x-\frac{\mathbf{m}}{N})^{\mathbf{n}} \\
&\leq \frac{2^d d^n}{n!}\left(\frac{1}{N}\right)^n + \frac{2^d}{n}\left(\left(\frac{d}{N}\right)^n + \cdots + \left(\frac{d}{N}\right)^1\left(\frac{d}{N}\right)^{n-1}\right) \\
&\leq 2^d\left(\frac{d}{N}\right)^n\left(1 + \frac{1}{n!}\right)
\end{aligned} \tag{13}$$

The second step follows $\psi_{\mathbf{m}}(\mathbf{x}) = 0$ when $\mathbf{x} \notin \text{supp}\psi_{\mathbf{m}}$. In the third step we turn the sum to multiplication, because for any $\mathbf{x}$ there are up to $2^d$ terms $\psi_{\mathbf{m}}(\mathbf{x})$ that are not equal to zero. The fourth step uses a Lagrange's form of the Taylor remainder. The fifth step follows different round

precision of $\beta_{\mathbf{m},\mathbf{n}}$ in different order and the fact that the number of terms with order $i$ is not greater than $d^i$.

We rewrite $f_2$ as

$$f_2(\mathbf{x}) = \sum_{\mathbf{m}\in\{0,\cdots,N\}^d} \sum_{\mathbf{n}:|\mathbf{n}|<n} \beta_{\mathbf{m},\mathbf{n}} f_{\mathbf{m},\mathbf{n}}(\mathbf{x}), \tag{14}$$

where

$$f_{\mathbf{m},\mathbf{n}}(\mathbf{x}) = \psi_{\mathbf{m}}\left(\mathbf{x} - \frac{\mathbf{m}}{N}\right)^{\mathbf{n}}. \tag{15}$$

Note that $\beta_{\mathbf{m},\mathbf{n}}$ is a constant and thus $f_2$ is a linear combination of at most $d^n(N+1)^d$ terms of $f_{\mathbf{m},\mathbf{n}}(\mathbf{x})$. Note that when $d=1$, the number of terms should be $n(N+1)^d$ instead; but for simplicity of presentation we loosely use the same expression as they are on the same order.

We define an approximation to $f_{\mathbf{m},\mathbf{n}}(\mathbf{x})$ as $f'_{\mathbf{m},\mathbf{n}}(\mathbf{x})$. The only difference between $f_{\mathbf{m},\mathbf{n}}(\mathbf{x})$ and $f'_{\mathbf{m},\mathbf{n}}(\mathbf{x})$ is that all multiplication operations are approximated by $\times'$ as discussed in Proposition 2. Consider that if we construct our function $\times'$ with $|\times'(x,y) - xy| < \epsilon_{\times'} = 2^{-2(r+1)}$, then

$$|\times'(x,y) - xz| \le |x(y-z)| + \epsilon_{\times'}. \tag{16}$$

Applying Equation (16) to $|f'_{\mathbf{m},\mathbf{n}}(\mathbf{x}) - f_{\mathbf{m},\mathbf{n}}(\mathbf{x})|$ repeatedly, we bound it to Equation (17).

$$\left| f'_{\mathbf{m},\mathbf{n}}(\mathbf{x}) - f_{\mathbf{m},\mathbf{n}}(\mathbf{x}) \right| =$$
$$\left| \times'\left(h(3Nx_1 - 3m_1), \cdots, \times'\left(h(3Nx_d - 3m_d), \times'\left(\left(x_{i_1} - \frac{m_1}{N}\right), \times'\left(\cdots, \left(x_{i_{|\mathbf{n}|}} - \frac{m_{|\mathbf{n}|}}{N}\right)\right)\right)\right)\right) \right.$$
$$\left. - \left(h(3Nx_1 - 3m_1)\left(h(3Nx_2 - 3m_2)\cdots\left(h(3Nx_d - 3m_d)\left(\left(x_{i_1} - \frac{m_{i_1}}{N}\right)\cdots\left(x_{i_{|\mathbf{n}|}} - \frac{m_{i_{|\mathbf{n}|}}}{N}\right)\right)\right)\right)\right) \right|$$
$$\le (d + |\mathbf{n}|)\epsilon_{\times'} \qquad i_1, \cdots, i_{|\mathbf{n}|} \in \{1, 2, \cdots, d\}$$
$$\tag{17}$$

Finally, we define our approximation to $f(\mathbf{x})$ as $f'(\mathbf{x})$:

$$f'(\mathbf{x}) \triangleq \sum_{\mathbf{m}\in\{0,\cdots,N\}^d} \sum_{\mathbf{n}:0<|\mathbf{n}|<n} \beta_{\mathbf{m},\mathbf{n}} f'_{\mathbf{m},\mathbf{n}}(\mathbf{x}). \tag{18}$$

Using Equation (17), we get the error bound of the approximation to $f_2(x)$ as in Equation (19).

$$|f'(\mathbf{x}) - f_2(\mathbf{x})| = \left| \sum_{\mathbf{m}\in\{0,\cdots,N\}^d} \sum_{\mathbf{n}:|\mathbf{n}|<n} \beta_{\mathbf{m},\mathbf{n}}\left(f'_{\mathbf{m},\mathbf{n}}(\mathbf{x}) - f_{\mathbf{m},\mathbf{n}}(\mathbf{x})\right) \right|$$
$$= \left| \sum_{\mathbf{m}:x\in\text{supp}\psi_{\mathbf{m}}} \sum_{\mathbf{n}:|\mathbf{n}|<n} \beta_{\mathbf{m},\mathbf{n}}\left(f'_{\mathbf{m},\mathbf{n}}(\mathbf{x}) - f_{\mathbf{m},\mathbf{n}}(\mathbf{x})\right) \right| \tag{19}$$
$$\le 2^d \max_{\mathbf{m}:x\in\text{supp}\psi_{\mathbf{m}}} \sum_{\mathbf{n}:|\mathbf{n}|<n} |f'_{\mathbf{m},\mathbf{n}}(\mathbf{x}) - f_{\mathbf{m},\mathbf{n}}(\mathbf{x})|$$
$$\le 2^d d^n (d + n - 1)\epsilon_{\times'}.$$

The second line follows again the support property and statement (i) of Proposition 2. The third line uses the bound $|\beta_{\mathbf{m},\mathbf{n}}| \le 1$. The fourth line is obtained by inserting Equation (17).

Then the final approximation error bound is as follows:

$$|f'(\mathbf{x}) - f(\mathbf{x})| \le |f'(\mathbf{x}) - f_2(\mathbf{x})| + |f(\mathbf{x}) - f_2(\mathbf{x})|$$
$$\le 2^d d^n (d + n - 1)\epsilon_{\times'} + 2^{d+1}\left(\frac{d}{N}\right)^n. \tag{20}$$

Using statement (ii) of Proposition 2 and choosing $r$ as $r = \frac{\log(6N^n(d+n-1))}{2} - 1$, the approximation error turns to

$$|f'(\mathbf{x}) - f(\mathbf{x})| \le 3 \cdot 2^d \left(\frac{d}{N}\right)^n. \tag{21}$$

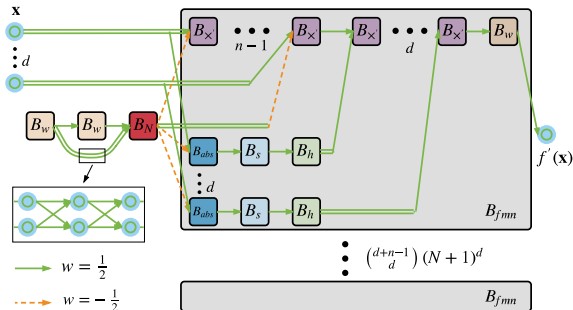

Figure 3: A qunatized ReLU network that implements $f'(x)$. The connections of all $B_{f_{mn}}$ are the same. Every connection from $B_N$ to other blocks has no greater than two weights.

Table 1: Block configuration for function-independent structure

| block | function | width | depth |
|---|---|---|---|
| $B_w$ | construct weights | $t$ | $\lambda\left(t^{\frac{1}{\lambda-1}}-1\right)+1$ |
| $B_N$ | construct $\frac{1}{N}, \cdots, \frac{N-1}{N}$ | $2N$ | $1$ |
| $B_{abs}$ | get absolute values | $4$ | $1$ |
| $B_s$ | scale by $3N$ | $4$ | $\log\frac{3N}{2}$ |
| $B_h$ | implement $h(x)$ | $12$ | $1$ |
| $B_{\times'}$ | implement $\times'(x,y)$ | $60$ | $3r+1$ |

Therefore, for any $f \in \mathcal{F}_{d,n}$ and $\epsilon \in (0,1)$, there is a ReLU network $f'$ that approximate $f$ with error bound $\epsilon$ if we choose $N \geq \left(3 \cdot 2^d d^n / \epsilon\right)^{\frac{1}{n}}$.

We now present the construction of the network for $f'(\mathbf{x})$. If every $f'_{\mathbf{m,n}}(\mathbf{x})$ can be computed by a sub-network, then $f'(\mathbf{x})$ is simply a weighted sum of all outputs of $f'_{\mathbf{m,n}}(\mathbf{x})$. By Proposition 3, we can implement the needed weights $\beta_{\mathbf{m,n}}$ by choosing $t = \log\frac{nN^n}{d^n}$. Then we simplify the task to constructing $f'_{\mathbf{m,n}}(\mathbf{x})$.

$\times'$ can be implemented as discussed in Proposition 2. For $h\left(\left(3N\left(x_i - \frac{m_i}{N}\right)\right)\right)$, noticing that $h(x) = h(-x)$, we can first compute $|x_i - \frac{m_i}{N}|$ as $\sigma(x_i - \frac{m_i}{N}) + \sigma(\frac{m_i}{N} - x_i)$ and then scale it to $3N\left(|x_i - \frac{m_i}{N}|\right)$. The implementation of $h(x)$ can thus be simplified as $1 - \sigma(x-1) + \sigma(x-2)$ since the input is nonnegative. Furthermore, by choosing $N$ as $cd^2$ where $c \in \mathbb{N}$ and $c > 1$, $\frac{1}{N}$ is an integral multiple of $\frac{1}{n}\left(\frac{d}{N}\right)^n$ if $n > 1$. When $n = 1$, $\frac{1}{N}$ is an integral multiple of $\left(\frac{1}{n}\left(\frac{d}{N}\right)^n\right)^2$. As discussed in Proposition 3, we build a weight construction network $B_w$ in the way that all integral multiples of the minimal precision can be obtained. Therefore, all $\frac{m_i}{N}$ can be obtained in the same way as $\beta_{\mathbf{m,n}}$, except that we need to concatenate two weight construction sub-networks.

Now we analyze the complexity of the network. The implementation of $f'(x)$ is shown in Figure 3. The function and size of blocks are listed in Table 1. Then we are able to obtain the complexity of the network. While we can write the complexity of the network in an explicit expression, here we use the $\mathcal{O}$ notation for clarity. Let $N_d, N_w, N_b$ be the depth, the number of weights, and the number of bits required respectively. The weight construction blocks $B_w$ have the highest order of number of weights and we have $N_w = \mathcal{O}\left(\lambda t^{\frac{1}{\lambda-1}+1} N^d\right)$. Meanwhile, we get $N_d = \mathcal{O}\left(\lambda t^{\frac{1}{\lambda-1}} + \log N\right)$. Inserting $t = \log\frac{nN^n}{d^n}$ and $N = \mathcal{O}\left((1/\epsilon)^{\frac{1}{n}}\right)$, we get $N_d = \mathcal{O}\left(\lambda \log^{\frac{1}{\lambda-1}}(1/\epsilon) + \log(1/\epsilon)\right)$ and $N_w = \mathcal{O}\left(\lambda \log^{\frac{1}{\lambda-1}+1}(1/\epsilon)(1/\epsilon)^{\frac{d}{n}}\right)$. Multiplying $N_w$ by $\log\lambda$, $N_b$ is obtained. This concludes the proof of Theorem 1. $\qquad\square$

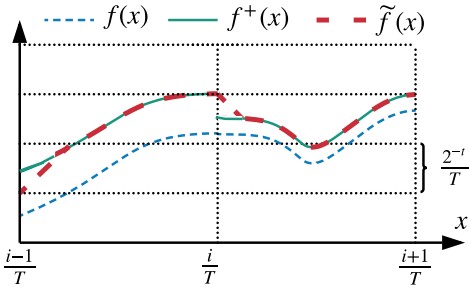

Figure 4: An example to illustrate the relationship between $f(x)$, $f^+$ and $\widetilde{f}(x)$.

## A.3 THE PROOF OF THEOREM 2

**Theorem 2.** *For any $f \in \mathcal{F}_{d,n}$, given weight maximum precision $\frac{1}{\lambda}$, there is a ReLU network with fixed structure that can approximate $f$ with any error $\epsilon \in (0, 1)$, such that (i) the depth is $\mathcal{O}\left(\log\left(1/\epsilon\right)\right)$; (ii) the number of weights is $\mathcal{O}\left(\left(\log\left(1/\epsilon\right) + \frac{\log^2(1/\epsilon)}{\log \lambda}\right)(1/\epsilon)^{\frac{d}{n}}\right)$; (iii) the number of bits needed to store the network is $\mathcal{O}\left(\left(\log(\lambda)\log\left(1/\epsilon\right) + \log^2\left(1/\epsilon\right)\right)(1/\epsilon)^{\frac{d}{n}}\right)$.*

With $\lambda$ distinct values, a linearly quantized network has a minimal resolution of $\frac{1}{\lambda}$. The proof for the approximability of linear quantization can be done in the same way as Theorem 1 except for a different sub-network for weight approximation. We still construct $W_c$ in Proposition 3 first and any weight value from $W_c$ can be obtained by multiply at most $\frac{t}{\log \lambda}$ weights. Thus the width and depth of the weight approximation network will be $t$ and $\frac{t}{\log \lambda} + 1$ respectively. Updating the $B_w$ in Table 1, we obtain the complexity accordingly.

## A.4 THE PROOF OF PROPOSITION 4

**Proposition 4.** *For any $f \in \mathcal{F}_{1,1}$, $t \in \mathbb{Z}^+$, and $T \in \mathbb{Z}^+$, there exists a function $\widetilde{f}(x)$ such that (i) $\widetilde{f}(x)$ is a continuous function with Lipschitz constant 1; (ii) $\widetilde{f}(\frac{i}{T}) = \left\lceil Tf\left(\frac{i}{T}\right)/2^{-t}\right\rceil \frac{2^{-t}}{T}$; (iii) $|\widetilde{f}(x) - f(x)| < \frac{2^{-t}}{T}$.*

*Proof.* We first divide $[0, 1]$ into $T$ intervals uniformly and define $f^+(x)$ as

$$f^+(x) \triangleq f(x) + \left(\left\lceil Tf\left(\frac{\lceil Tx\rceil}{T}\right)/2^{-t}\right\rceil \frac{2^{-t}}{T} - f\left(\frac{\lceil Tx\rceil}{T}\right)\right). \tag{22}$$

Note that $f^+(\frac{i}{T}) = \left\lceil Tf\left(\frac{i}{T}\right)/2^{-t}\right\rceil \frac{2^{-t}}{T}$ and $\frac{\mathrm{d}f^+(x)}{\mathrm{d}x} = \frac{\mathrm{d}f(x)}{\mathrm{d}x}$ on $(\frac{i}{T}, \frac{i+1}{T})$ where $, i = 1, 2, \cdots, T$. Then, we define $\widetilde{f}(x)$:

$$\widetilde{f}(0) \triangleq \left\lceil \frac{Tf(0)}{2^{-t}}\right\rceil \frac{2^{-t}}{T}, \tag{23}$$

$$\frac{\mathrm{d}\widetilde{f}(x)}{\mathrm{d}x} \triangleq \begin{cases} \operatorname{sgn}(f^+(x) - \widetilde{f}(x)) & \widetilde{f}(x) \neq f^+(x) \\ \frac{\mathrm{d}f(x)}{\mathrm{d}x} & \widetilde{f}(x) = f^+(x). \end{cases} \tag{24}$$

$f^+(x)$ is parallel with $f(x)$ on any given interval $(\frac{i}{T}, \frac{i+1}{T}]$ and is shifted to let $f^+(\frac{i}{T})$ be an integral multiple of $\frac{2^{-t}}{T}$. $\widetilde{f}(\frac{i}{T})$ is $f(\frac{i}{T})$ rounded up to an integral multiple of $\frac{2^{-t}}{T}$. Meanwhile, $\widetilde{f}(x)$ approaches $f(x)$ with fixed slope whenever they are not equal. An example is shown in Figure 4.

Statement (i) follows the definition of $\widetilde{f}(x)$ directly. We give a proof for statement (ii) by contradiction and induction. If $\widetilde{f}(\frac{i}{T}) = f^+(\frac{i}{T})$, then $\widetilde{f}(x) = f^+(x)$ on $(\frac{i}{T}, \frac{i+1}{T}]$. The proof idea for $\widetilde{f}(\frac{i}{T}) > f^+(\frac{i}{T})$ and $\widetilde{f}(\frac{i}{T}) < f^+(\frac{i}{T})$ are basically the same thus we present the proof of the latter for brevity. We first assume $\widetilde{f}(\frac{i}{T}) = \left\lceil Tf\left(\frac{i}{T}\right)/2^{-t}\right\rceil \frac{2^{-t}}{T}$, which is satisfied by Equation (23) when

$i = 0$. Since $\widetilde{f}(x) = f^+(x)$ after the first intersection on $(\frac{i}{T}, \frac{i+1}{T}]$, if $\widetilde{f}(x)$ and $f^+(x)$ have no intersection on $(\frac{i}{T}, \frac{i+1}{T}]$, then $\widetilde{f}(\frac{i+1}{T}) < f^+(\frac{i+1}{T})$ because $\widetilde{f}(\frac{i}{T}) < f^+(\frac{i}{T})$. Meanwhile, we have $\frac{\mathrm{d}\widetilde{f}(x)}{\mathrm{d}x} = 1$, and

$$
\begin{aligned}
\widetilde{f}\left(\frac{i+1}{T}\right) &= \widetilde{f}\left(\frac{i}{T}\right) + \int_{\frac{i}{T}}^{\frac{i+1}{T}} \frac{\mathrm{d}\widetilde{f}(x)}{\mathrm{d}x}dx \\
&= \left\lceil Tf\left(\frac{i}{T}\right)/2^{-t}\right\rceil \frac{2^{-t}}{T} + \frac{1}{T} \\
&= \left\lceil T\left(f\left(\frac{i}{T}\right) + \frac{1}{T}\right)/2^{-t}\right\rceil \frac{2^{-t}}{T} \\
&\geq \left\lceil Tf\left(\frac{i+1}{T}\right)/2^{-t}\right\rceil \frac{2^{-t}}{T} \\
&\geq f^+\left(\frac{i+1}{T}\right).
\end{aligned}
\tag{25}
$$

This contradicts the fact that $\widetilde{f}(\frac{i+1}{T}) < f^+(\frac{i+1}{T})$. Thus $\widetilde{f}(x)$ and $f^+(x)$ intersect on $(0, \frac{i}{T}]$ and in turn guarantee that $\widetilde{f}(\frac{i+1}{T}) = \left\lceil Tf\left(\frac{i+1}{T}\right)/2^{-t}\right\rceil \frac{2^{-t}}{T}$. By induction, we prove that $\widetilde{f}(x)$ and $f^+(x)$ intersect on every interval $(\frac{i}{T}, \frac{i+1}{T}]$. This implies statement (ii).

Now we prove the statement (iii). Note that we have $0 \leq \widetilde{f}(\frac{i}{T}) - f(\frac{i}{T}) \leq \frac{2^{-t}}{T}$ by statement (ii). In every interval $(\frac{i}{T}, \frac{i+1}{T})$, $\widetilde{f}(x) = f^+(x)$ after the their intersection. Therefore we have $0 \leq |f^+(x) - f(x)| \leq \frac{2^{-t}}{T}$ by Equation (22). Before the intersection, if $\widetilde{f}(0) < f^+(0)$, $\frac{\mathrm{d}(\widetilde{f}(x)-f^+(x))}{\mathrm{d}x} \geq 0$. Since $\frac{\mathrm{d}f^+(x)}{\mathrm{d}x} = \frac{\mathrm{d}f(x)}{\mathrm{d}x}$ on $(\frac{i}{T}, \frac{i+1}{T})$, we have $\frac{\mathrm{d}(\widetilde{f}(x)-f(x))}{\mathrm{d}x} \geq 0$, thus $0 \leq \widetilde{f}(\frac{i}{T}) - f(\frac{i}{T}) \leq \widetilde{f}(x) - f(x) \leq f^+(x) - f(x) \leq \frac{2^{-t}}{T}$. If $\widetilde{f}(\frac{i}{T}) \geq f^+(\frac{i}{T})$, apply the same logic and we obtain $0 \leq f^+(x) - f(x) \leq \widetilde{f}(x) - f(x) \leq \widetilde{f}(\frac{i}{T}) - f(\frac{i}{T}) \leq \frac{2^{-t}}{T}$. This implies statement (iii) and concludes the proof. $\qquad\square$

## A.5 The proof of Theorem 3

**Theorem 3.** *For any $f \in \mathcal{F}_{1,1}$, given $\lambda$ distinct weights, there is a ReLU network with function-dependent structure that can approximate $f$ with any error $\epsilon \in (0, 1)$, such that (i) the depth is $\mathcal{O}\left(\lambda \left(\log\log\left(1/\epsilon\right)\right)^{\frac{1}{\lambda-1}} + \log\left(1/\epsilon\right)\right)$; (ii) the number of weights is $\mathcal{O}\left(\lambda \left(\log\log\left(1/\epsilon\right)\right)^{\frac{1}{\lambda-1}+1} + \left(1/\epsilon\right)\right)$ (iii) the number of bits needed to store the network is $\mathcal{O}\left(\log\lambda \left(\lambda \left(\log\log\left(1/\epsilon\right)\right)^{\frac{1}{\lambda-1}+1} + \left(1/\epsilon\right)\right)\right)$.*

*Proof.* We first transform $f$ to $f''$ with Proposition 4. Then we apply the interpolation and cached function method from [35] while using the weight construction method described in Proposition 3. Denoting the output of the network as $f''(x)$, we have $|f(x) - f''(x)| = |f(x) - \widetilde{f}(x)| + |\widetilde{f}(x) - f''(x)| \leq \epsilon$ by choosing the hyper-parameters as $m = \lceil\frac{1}{2}\log\left(1/\epsilon\right)\rceil$, $t = \lceil\log m\rceil$, $\delta = \frac{1}{8m}$, $T = \lceil\frac{8}{\epsilon\log(1/\epsilon)}\rceil$.

The approximation network is shown in Figure 5. The sizes of blocks are given in Table 2 where $f^T$ is the uniform linear interpolation function of $f$ with $T-1$ breakpoints, $f^*$ is the sum of the selected cached functions, $\Phi(x)$ is a filtering function. The inputs connections to $B_{f^*}$ and the connections inside $B_m$ have higher order to the number of weights than others. Then the complexity can be obtained accordingly. $\qquad\square$

## A.6 The proof of Theorem 4

**Theorem 4.** *For any $f \in \mathcal{F}_{1,1}$, given weight maximum precision $\frac{1}{\lambda}$, there is a ReLU network with function-dependent structure that can approximate $f$ with any error $\epsilon \in (0, 1)$, such that (i) the*

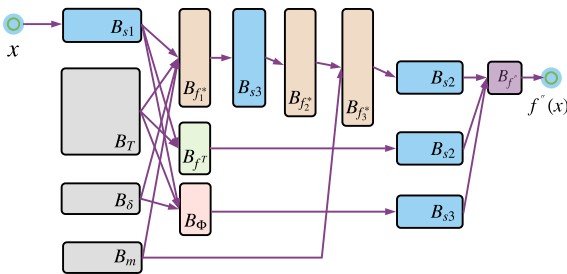

Figure 5: A qunatized ReLU network that implements $f''(x)$. Illustration only, details are omitted. For example, arrows between blocks can represent more than one connection in the network and there are short-cut connections that allow scaling only part of the inputs.

Table 2: Block configuration for function-dependent structure. Terms with lower order are omitted.

| block | function | width | depth |
|---|---|---|---|
| $B_T$ | construct $1, 2, \cdots, T$ | $\log T$ | $\log T$ |
| $B_\delta$ | construct $\delta$ | 1 | $\lambda(t^{\frac{1}{\lambda-1}})$ |
| $B_m$ | construct $\frac{1}{m}, \cdots, \frac{m-1}{m}$ | $t$ | $\lambda(t^{\frac{1}{\lambda-1}})$ |
| $B_{f^T}$ | implement $f^T(x)$ | $T$ | 1 |
| $B_\Phi$ | implement $\Phi(x)$ | $4T$ | 1 |
| $B_{s1}$ | scale by $T$ | 4 | $\log T$ |
| $B_{s2}$ | scale by $\frac{1}{T}$ | 1 | $\log T$ |
| $B_{s3}$ | scale by $\frac{1}{\delta}$ | 4 | $\log m$ |
| $B_{f_1^*}$ | first layer of $f^*$ | $T$ | 1 |
| $B_{f_2^*}$ | second layer of $f^*$ | $3^m$ | 1 |
| $B_{f_3^*}$ | third layer of $f^*$ | $m3^m$ | 1 |
| $B_{f''}$ | implement $f''(x)$ | 4 | 1 |

depth is $\mathcal{O}\left(\log\left(1/\epsilon\right)\right)$; (ii) the number of weights is $\mathcal{O}\left(1/\epsilon\right)$; (iii) the number of bits needed to store the network is $\mathcal{O}\left(\log(\lambda)/\epsilon\right)$.

*Proof.* The proof for the approximability of linear quantization can be done in the same way as Theorem 3 except for a different sub-network for weight approximation. We still construct $W_c$ in Proposition 3 first and any weight value from $W_c$ can be obtained by multiply at most $\frac{t}{\log \lambda}$ weights. Thus the width and depth of the weight approximation network will be $t$ and $\frac{t}{\log \lambda} + 1$ respectively. Updating the $B_\delta$ and $B_m$ in Table 2, we obtain the complexity accordingly. □

## B  THE EXISTENCE OF AN OPTIMAL BIT-WIDTH

In this section, we prove the statement in Section 6 that there exists one and only one local minimum (hence global minimum) for $M(\lambda)$ in the range of $[2, \infty)$ whenever $\epsilon < \frac{1}{2}$. Denote $\log\left(3n2^d/\epsilon\right)$ as $\theta_2$ and we get the derivative of $M(\lambda)$ as:

$$\frac{dM}{d\lambda} = \theta_2^{\frac{\lambda}{\lambda-1}} \left( \log(\lambda) + \frac{1}{\ln 2} - \ln(\theta_2)\frac{\lambda \log(\lambda)}{(\lambda-1)^2} \right) \tag{26}$$

Let $M_s(\lambda) = \log(\lambda) + \frac{1}{\ln 2} - \ln(\theta_2)\frac{\lambda \log(\lambda)}{(\lambda-1)^2}$. Since $\theta_2^{\frac{\lambda}{\lambda-1}} > 0$, we have $\text{sgn}(M_s) = \text{sgn}\left(\frac{dM}{d\lambda}\right)$.

We have

$$\frac{dM_s}{d\lambda} = \frac{1}{\lambda} + \frac{1 - \lambda + \log(\lambda) + \lambda\log(\lambda)}{(\lambda-1)^3} > 0, \qquad \forall \lambda \geq 2 \tag{27}$$

$M_s(2) = 1 + \frac{1}{\ln 2} - 2\ln(\theta_2) < 0$ given $\epsilon < \frac{1}{2}$, and $\lim_{\lambda \to \infty} M_s(\lambda) = \infty$. It is clear that there exist a $\lambda_{opt}$ determined by $\theta_2$ such that $\text{sgn}(M_s(\lambda)) = -1$ on $[2, \lambda_{opt})$ and $\text{sgn}(M_s(\lambda)) = 1$ on $(\lambda_{opt}, \infty)$.

Remember that $\mathrm{sgn}(M_s) = \mathrm{sgn}\left(\dfrac{dM}{d\lambda}\right)$, then $\lambda = \lambda_{opt}$ is the one and only one local minimum of $M(\lambda)$ on $[2, \infty)$.

