# OpenReview forum: "On the Universal Approximability and Complexity Bounds of Quantized ReLU Neural Networks"
_ICLR.cc/2019/Conference_

### Official Review · AnonReviewer1 · 2018-10-29
**A very interesting and rather clear paper on quantized ReLU neural networks**

**Rating:** 8
**Confidence:** 3

**Review:**

The authors propose in this paper a series of results on the approximation capabilities of neural networks based on ReLU using quantized weights. Results include upper bounds on the depth and on the number of weights needed to reach a certain approximation level given the number of distinct weights usable. The paper is clear and as far as I know the results are both new and significant. My only negative remark is about the appendix that could be clearer. In particular, I think that figure 2 obscures the proof of Proposition 1 rather than the contrary. I think it might be much clearer to give an explicit neural network approximation of x^2 for say r=2, for instance.

---

> ### Author Response · Authors · 2018-11-12
> **Response to Review #1 titled “A very interesting and rather clear paper on quantized ReLU neural networks”**
>
> We thank you for your comments and support. We appreciate that you value our new and significant results in this new direction. We have revised Figure 2 in the Appendix in connection with the proof of Proposition 1 and added some detailed descriptions as below. A few other revisions are also made in the Appendix to improve the overall presentation. Please check the new version for details.
> “
> Note that a straightforward implementation will have to scale $g^{\circ i}(x)$ separately (multiply by different numbers of $\frac{1}{2}$) before subtracting them from $x$ because each $g^{\circ i}(x)$ have a different coefficient. Then the width of the network will be $\Theta(r)$. Here we use a ``pre-scale'' method to reduce the network width from $\Theta(r)$ to a constant. The network constructed is shown in Figure 2. The one-layer sub-network that implements $g(x)$ and the one-layer sub-network that scales the input by $4$ are denoted as $B_g$ and $B_m$ respectively. Some units are copied to compensate the scaling caused by $\frac{1}{2}$. The intermediate results $g^{\circ i}(x)$ are computed by the concatenation of $B_g$ at the $(i+1)$-th layer. The first $B_m$ takes $x$ as input and multiply it by $4$. The output of $i$-th $B_m$ is subtracted by $g^{\circ i}(x)$ and then fed to the next $B_m$ to be multiplied by $4$ again. There are $r$ layers of $B_m$ and all $g^{\circ i}(x)$ are scaled by $2^{2(r-i)}$ respectively. As a result, we obtain $2^{2r}x-\sum_{i=1}^{r}2^{2(r-i)}g^{\circ i}(x)$ after the last $B_m$. Then it is scaled by $2^{-2r}$ in the later $2r$ layers to get $f^r_s(x)$. In this way, we make all $g^{\circ i}(x)$ sharing the same scaling link and a constant width can be achieved.
> "

---

### Official Review · AnonReviewer3 · 2018-11-01
**Reasonable paper on an interesting topic**

**Rating:** 6
**Confidence:** 3

**Review:**

The paper deals with the expressibility of quantized neural network, meaning where all weights come from a finite and small sized set. It proves that functions satisfying standard assumptions can be represented by quantized ReLU networks with certain size bounds, which are comparable to the bounds available in prior literature for general ReLU networks, with an overhead that depends on the level of quantization and on the target error.

The proofs generally go by simulating non-quantized ReLU networks with quantized ones, by means of replacing their basic operations with small quantized networks ("sub-networks") that simulate those same operations with a small error. Then the upper bounds follow from known results on function approximation with (non-quantized) ReLU networks, with the overhead incurred by introducing the sub-networks.
Notably, this approach means that the topology of the network changes. As such it not compatible with quantizing the weights of a given network structure, which is the more common scenario, but rather with choosing the network structure under a given level of quantization. This issue is discussed directly and clearly in the paper.

Overall, while the paper is technically quite simple, it forms an interesting study and blends well into recent literature on an important topic. It is also well written and clear to follow.

---

> ### Author Response · Authors · 2018-11-12
> **Response to Review #3 titled “Reasonable paper on an interesting topic”**
>
> We thank you for your time and thoughtful review. In fact, the change of the topology with the level of quantization is not an issue but part of the construction of the proof. Note that the goal of this work is to provide a theoretical proof on the expressive power of quantized neural networks without any assumptions on how we obtain the quantized networks. By allowing the networks to have different topologies, we are able to mathematically prove how we can use constructed quantized networks to approximate the same target function within any given error bound. With this flexibility, we are able to obtain the bound on the number of parameters and make a fair comparison between quantized networks and unquantized networks. Since there is no previous work on the theoretical expressive power of quantized neural networks, we consider our work as a good first attempt. Of course, a natural research question is whether we can extend the theoretical result to a given network (but not a given target function as studied in this paper). We would like to explore such a question in our future research.

---

### Official Review · AnonReviewer2 · 2018-11-04
**Review for On the Universal Approximability and Complexity Bounds of Quantized ReLU Neural Networks**

**Rating:** 7
**Confidence:** 3

**Review:**

This paper studies the expressive power of quantized ReLU networks from a theoretical point of view. This is well-motivated by the recent success of using quantized neural networks as a compression technique. This paper considers both linear quantization and non-linear quantization, both function independent network structures and function dependent network structures. The obtained results show that the number of weights need by a quantized network is no more than polylog factors times that of a unquantized network. This justifies the use of quantized neural networks as a compression technique.

Overall, this paper is well-written and sheds light on a well-motivated problem, makes important progress in understanding the full power of quantized neural networks as a compression technique. I didn’t check all details of the proof, but the structure of the proof and several key constructions seem correct to me. I would recommend acceptance.

The presentation can be improved by having a formal definition of linear quantized networks and non-linear quantized networks, function-independent structure and function-dependent structure in Section 3 to make the discussion mathematically rigorous. Also, some of the ideas/constructions seem to follow (Yarotsky, 2017). It seems to be a good idea to have a paragraph in the introduction to have a more detailed comparison with (Yarotsky, 2017), highlighting the difference of the constructions, the difficulties that the authors overcame when deriving the bounds, etc.

Minor Comment: First paragraph of page 2: extra space after ``to prove the universal approximability’’.

---

> ### Author Response · Authors · 2018-11-12
> **Response to Review #2 titled “Review for On the Universal Approximability and Complexity Bounds of Quantized ReLU Neural Networks”**
>
> Thank you for your positive and constructive feedback. We have made the revision to further improve the presentation following your suggestions. Please check the new version for details.
>
> We made the definition of two types of quantization, linear quantization and nonlinear quantization, and two types of structure, function-dependent structure and function-independent structure, more formal as below and moved them to Section 3 “Models and Assumptions”.
>
> Here are the paragraphs related to the definition of linear vs nonlinear quantization:
> “
> We denote the finite number of distinct weight values as $\lambda$ ($ \lambda \in \mathbb{Z}^{+}$ and $\lambda \geq 2$), for both linear and nonlinear quantization. For linear quantization, without loss of generality, we assume the finite number of distinct weight values are given as $ \{-1, \frac{1}{\lambda},\frac{2}{\lambda},\dots,\frac{\lambda-1}{\lambda}\}$, where $\{\frac{1}{\lambda},\frac{2}{\lambda},\dots,\frac{\lambda-1}{\lambda}\}$ are uniformly spaced (hence called ``linear’’)  in $(0,1)$ and $-1$ is used to obtain the negative weight values. For nonlinear quantization, we assume the finite number of distinct weight values are not constrained to any specific values, i.e., they can take any values as needed.
> ”
>
> Here are the paragraphs related to the definition of the function-dependent vs independent structures.
> “
> When constructing the network to approximate any target function $f$, we consider two scenarios for deriving the bounds.  The first scenario is called function-dependent structure, where the constructed network topology and their associated weights are all affected by the choice of the target function. In contrast, the second scenario is called function-independent structure, where the constructed network topology is independent of the choice of the target function in $ f\in\mathcal{F}_{d,n}$ with a given $\epsilon$. The principle behind these design choices (the network topology constructions and the choice of weights) is to achieve a tight upper bound as much as possible.
> ”
>
> We added more discussion of the difference between our work and that of (Yarotsky, 2017), and moved that discussion from Section 2 “Related Works” to Section 1 “Introduction”. Details are quoted as follows:
> “
> We follow the idea from (Yarotsky, 2017) to prove the complexity bound by constructing a network, but with new and additional construction components essential for quantized networks. Specifically, given the number of distinct weight values $\lambda$ and a target function $f$, we construct a network that can approximate $f$ with an arbitrarily small error bound $\epsilon$ to prove the universal approximability. The memory size of this network then naturally serves as an upper bound for the minimal network size.
> The high-level idea of our approach is to replace basic units in an unquantized network with quantized sub-networks that approximate these basic units. For example, we can approximate a connection with any weight in an unquantized network by a quantized sub-network that only uses a finite number of given weight values. Even though the approximation of any single unit can be made arbitrarily accurate in principle with unlimited resources (such as increased network depth), in practice, there exists some inevitable residual error at every approximation, all of which could propagate throughout the entire network. The challenge becomes, however, how to mathematically prove that we can still achieve the end-to-end arbitrary small error bound even if these unavoidable residual errors caused by quantization can be propagated throughout the entire network. This paper finds a solution to solve the above challenge. In doing so, we have to propose a number of new ideas to solve related challenges, including judiciously choosing the proper finite weight values, constructing the approximation sub-networks as efficient as possible (to have a tight upper bound), and striking a good balance among the complexities of different approximation steps.
> ”

---

### Author Response · Authors · 2018-11-12
**Submission Updated**

We thank all reviewers for their time and valuable comments. We are grateful that reviewers found this paper interesting, important, and clear. We have carefully revised the paper following the reviewers’ suggestions to further improve the presentation and have updated the submission. The revisions we made include the following:

1. We made the definition of two types of quantization, linear quantization and nonlinear quantization, and two types of structure, function-dependent structure and function-independent structure, more formal and moved them to Section 3 “Models and Assumptions”.
2. We added more discussion of the difference between our work and that of (Yarotsky, 2017), and moved that discussion from Section 2 “Related Works” to Section 1 “Introduction”.
3. We revised Figure 2 in the Appendix in connection with the proof of Proposition 1 and added some detailed descriptions.
4. A few minor fixes like re-organizing sentences and correcting typos.

---

### Meta-Review · Area_Chair1 · 2018-12-16
**Good motivation and new theoretical insights**

**Confidence:** 4
**Recommendation:** Accept (Poster)

**Metareview:**

This paper addresses a well motivated problem and provides new insight on the theoretical analysis of representational power in quantized networks. The results contribute towards a better understanding of quantized networks in a way that has not been treated in the past.

The most moderate rating (marginally above acceptance threshold) explains that while the paper is technically quite simple, it gives an interesting study and blends well into recent literature on an important topic.

A criticism is that the approach uses modules to approximate the basic operations of non quantized networks. As such it not compatible with quantizing the weights of a given network structure, but rather with choosing the network structure under a given level of quantization. However, reviewers consider that this issue is discussed directly and clearly in the paper.

The reviewers report to be only fairly confident about their assessment, but they all give a positive or very positive evaluation of the paper.